# Essential Oils as Natural Sources of Fragrance Compounds for Cosmetics and Cosmeceuticals

**DOI:** 10.3390/molecules26030666

**Published:** 2021-01-27

**Authors:** Jugreet B. Sharmeen, Fawzi M. Mahomoodally, Gokhan Zengin, Filippo Maggi

**Affiliations:** 1Department of Health Sciences, Faculty of Medicine and Health Sciences, University of Mauritius, Réduit 80837, Mauritius; sharmeenjugs@gmail.com; 2Department for Management of Science and Technology Development, Ton Duc Thang University, Ho Chi Minh City 758307, Vietnam; 3Faculty of Applied Sciences, Ton Duc Thang University, Ho Chi Minh City 758307, Vietnam; 4Physiology and Biochemistry Research Laboratory, Department of Biology, Science Faculty, Selcuk University, Konya 42130, Turkey; gokhanzengin@selcuk.edu.tr; 5School of Pharmacy, University of Camerino, via Sant’Agostino 1, 62032 Camerino, Italy; filippo.maggi@unicam.it

**Keywords:** essential oils, natural fragrances, cosmetics

## Abstract

Fragrance is an integral part of cosmetic products and is often regarded as an overriding factor in the selection of cosmetics among consumers. Fragrances also play a considerable role in masking undesirable smells arising from fatty acids, oils and surfactants that are commonly used in cosmetic formulations. Essential oils are vital assets in the cosmetic industry, as along with imparting pleasant aromas in different products, they are able to act as preservatives and active agents and, simultaneously, offer various benefits to the skin. Moreover, the stimulating demand for natural ingredients has contributed massively to a renewed interest in cosmetic and wellness industries in plant derivatives, especially essential oils. This has led popular cosmetic companies to endorse natural fragrances and opt for minimally processed natural ingredients, given the potentially adverse health risks associated with artificial fragrance chemicals, which are major elements of cosmetics. Among the high-valued essential oils used as fragrances are citrus, lavender, eucalyptus, tea tree and other floral oils, among others, while linalool, geraniol, limonene, citronellol, and citral are much-appreciated fragrance components used in different cosmetics. Thus, this review aimed to highlight the enormous versatility of essential oils as significant sources of natural fragrances in cosmetics and cosmeceuticals. Moreover, a special focus will be laid on the different aspects related to essential oils such as their sources, market demand, chemistry, fragrance classification, aroma profile, authenticity and safety.

## 1. Introduction

The use of fragrance is ubiquitous and is a global human phenomenon. Over the course of time, countless numbers of flavors and fragrances have found their way into everyday life, notably into foods, beverages and confectionery items; into personal care products (soaps, toothpastes, mouthwashes, deodorants, bath lotions and shampoos), perfumes, and other cosmetics as well as pharmaceutical formulations. Indeed, flavors and aromas are added to make such products more attractive or to mask the taste or smell of less pleasant ones [1]. In recent times, green consumerism and the resurgence of the use of “naturals” have given a fresh impetus to the development of plant-based products, especially in beauty and wellness industries [2].

Fragrances play a particularly important role in increasing the attractiveness of cosmetics. Pleasant smells influence the comfortability and the effect of the products and also impact significantly on the overall evaluation of cosmetics. Therefore, along with the shape and design of the container, the smell is also one of the characteristics of cosmetics that the consumer experiences [3] and look forward to in the selection of those items. Among natural fragrances, essential oils, which are complex mixtures of terpenes and other aromatic or aliphatic compounds, produced as secondary metabolites in specialized secretory tissues of aromatic plants [4], are the most popular. In nature, essential oils play very important roles in plant defense and signaling processes. For instance, they are involved in plant defense against insects, herbivores, and microorganisms, including attraction of pollinating insects and fruit-dispersing animals, water regulation and allelopathic interactions [5,6,7]. Once the plants have been harvested, the essential oil is extracted by means of steam distillation, expression (physical crushing of essential oil glands situated in fruit rinds or outermost waxy layer of fruit’s peel) [8], microwave-assisted extraction, solvent extraction, or enfleurage (transfer of the essential oil from flower petals to fat). Of these methods, steam distillation is the most common overall, while expression is most commonly used to obtain essential oils from the peels of citrus fruits such as orange, lemon, bergamot, etc.) [9].

Indeed, the volatile nature of essential oils makes them likely to be useful as fragrances but does not preclude other functions for them in cosmetics [10]. In fact, their intended use has been for a long time principally industrial, whereby standardization of fragrance requiring the blending of essential oils from different plants with the aim of obtaining a specific scent has been a common practice [11]. It has been reported that 300 essential oils out of about 3000 plant species are commercially available in the flavors and fragrances market [12]. The essential oils produced predominantly for industrial purposes are those from orange, corn mint, citronella, eucalyptus, peppermint, and lemon [13].

Essential oils are widely incorporated into modern skincare products because of their complexity of active compounds, strongly fragrant properties and natural marketing image. Moreover, over the years, they have shown several scientifically proven cosmetic properties. For instance, their importance in cosmetic preparations as natural preservative agents, owing to their antimicrobial properties, alone or in combination with other preservatives is established [14], whereby they assure protection against bacteria and fungi. Hence conveniently, added essential oils are able to enhance the dermato-cosmetic properties of the final product, not only by protecting against microbial infections but also contributing to the preservation of the cosmetic formulation [15,16]. Interestingly, they are also used to induce additional benefits to the skin such as anti-acne, anti-aging, skin lightening, and sun protection, among others, in cosmeceutical products, thus rendering them highly valuable to cosmetic industries [17,18,19,20,21,22].

Many industries use synthetic fragrances that are developed in a laboratory to mimic the aromatic and chemical constituents of natural, plant-based oils that are more expensive to manufacture. Nevertheless, synthetic fragrances may not contain the beneficial aspects of natural plant-based essential oils and could even be unsafe for human applications [23]. For instance, chemicals found in man-made fragrances include phthalates, which are endocrine disruptors [24] and known carcinogens such as benzene derivatives [25]. On the other hand, the global natural fragrances market is witnessing a high growth due to increasing usage of natural fragrances such as essential oils over synthetic fragrances as a result of their associated numerous health benefits such as in aromatherapy which is expected to drive the growth of the market in the coming years. For instance, the global essential oils market size was found to exceed USD 7.51 billion in 2018 and is expected to grow at greater than 9% compound annual growth rate (CAGR) between 2019 and 2026 [26]. Moreover, famous cosmetic brands use various essential oils in cosmetics, and the most expensive perfumes contain pure essential oils, which give them special and unique scents and make them unique as exclusive fragrances [27]. However, although the effectiveness of essential oils in health and beauty is widely acknowledged, they may not be completely free from contraindications and allergic effects [28].

Indeed, large-scale harvesting of aromatic plants for commercial purposes can cause species loss, exposing them to the danger of extinction. It is, therefore, vital that systematic cultivation of these plants be introduced in order to conserve biodiversity and protect endangered species [29]. Sustainable extraction of natural products, making use of innovative technologies, process intensification and agro-based solvents constitute the way out in the development of eco-conceived fragrant ingredients covering every olfactory family without putting biodiversity at risk any further [30].

Hence, this review has aimed to provide a spotlight on the versatility of these botanical essences as an explicit source of natural fragrances commonly used in cosmetic and cosmeceutical products. Additionally, other pertinent aspects related to essential oils, such as their sources, market demand, chemistry, fragrance classifications, aroma profiles, as well as their authenticity and safety, have been briefly discussed.

## 2. Sources of Essential Oils

There are 400,000 known plant species of both aromatic and medicinal plants, of which about 2000 species come from nearly 60 botanical families of essential oils bearing plants [31]. The plant families composed of species yielding the majority of the most economically important essential oils are not limited to one specialized taxonomic group but can be found distributed among all plant classes [32] (Table 1).

Several plants contain essential oils; however, parts of plants which serve as the major source of essential oil, can be different. These include flowers and inflorescences (e.g., chamomile, lavender, rose, ylang-ylang), leaves (e.g., basil, laurel, lemongrass, peppermint, rosemary), fruits (black pepper, nutmeg), peel (orange, bergamot, lemon, tangerine), seeds (anise, cumin, cardamom, fennel), berries (allspice, juniper), bark (cinnamon, cassia, sassafras), wood (cedarwood, camphor, sandalwood), root and rhizomes (ginger, vetiver, turmeric), and resin (myrrh, frankincense) [33].

## 3. Demand for Essential Oils in Cosmetic Industries

Essential oils have been widely used all over the world, and their use is growing due to the strong demand for pure natural ingredients in various fields. Thus, large quantities of essential oils are produced globally to fuel the industries of fragrances and flavors, cosmetics, as well as phytomedicine and aromatherapy [34,35].

Demand comes mostly from the following markets: food and beverage (35%), fragrances, cosmetics and aromatherapy (29%), household (16%), and pharmaceutical (15%) [13]. As consumers have become increasingly conscious of the health benefits of essential oils, inclinations for foods and beverages containing these volatile oils as additives have developed. The global essential oils market has also been led by the growth in demand for organic and natural hygienic products owing to increasing awareness of health problems among consumers. Moreover, natural flavors and fragrances demand in perfumes, cosmetics, thermal and relaxation applications are likely to stimulate the demand for essential oils. Essential oils and oleoresins are not merely used in the food processing and industrial seasoning sectors but are also vital in the perfume and flavoring industries. Globally, operating flavors and fragrance manufacturers are among the main buyers of essential oils. Their sales present an indication of developments in their market and subsequent demand for essential oils [13].

Particularly in cosmeceutical industries, essential oils such as that from orange has garnered maximum revenue as it offers multiple health remedies such as skin elasticity, firmness and treats scars, acne, and stretch marks. Lemon and orange essential oils also have antiseptic properties, which make them ideal ingredients in skin and hair care. Furthermore, the antimicrobial properties of these oils have opened new avenues for further business growth [26].

Some essential oils are produced on a very large-scale (e.g., in 2008, production of orange oils was ~51,000 tons, corn mint oils ~32,000 tons, and lemon oils ~9200 tons). Essential oils obtained from a variety of fruits from the genus *Citrus* are the most popular natural essential oils and account for the largest part of commercial natural flavors and fragrances [36]. The production of some others at a much smaller scale due to their rarity is, however, traded at very elevated prices [e.g., agarwood oil (6000–11,000 €/kg), iris (6200–100,000 €/kg depending on the concentration of irones), or rose oil (6000–10,000 €/kg)]. These prices vary and may be related to the scarcity of the raw materials, harvesting issues, climate dependence, or yield of extraction. Essential oil industries cumulative sales represented several billion US$ in 2008 [34,37].

Europe accounts for the greatest share of the global essential oils market, with the Asia Pacific region and North America tying for second place. Essential oils play a key role in the manufacturing of scented candles, oils, and other products. In 2017, the essential oils industry in the United States witnessed a boost in revenue of nearly 10% and was expected to reach around 7.3 billion dollars in market value by 2024. The essential oils that make up the largest share of the US market are orange, corn mint, and eucalyptus essential oils. Moreover, as a country with a rich history of perfume manufacturing, France is also the top exporter of essential oils worldwide. The export value of essential oils and resinoids from the US, in recent years, has grown by nearly two billion US dollars. Moreover, some countries that cannot produce certain varieties of essential oils in huge quantities to satisfy the local demands must import them from other countries. For instance, Lebanon imported the most essential oils of any country globally, at around 13.75 billion US dollars in 2017, while Northern European countries such as Germany, the Netherlands, and the United Kingdom, accounted for almost 80% of the orange oil imported to Europe [38].

## 4. Chemistry of Essential Oils

Although the term “aromatic” in modern usage describes the quality of giving off an aroma that is either pleasant or odious to the nose, an aromatic compound or moiety, in the language of chemistry, has a chemical arrangement that results in delocalization of electrons, producing greater molecular stability. Thus, essential oils may be a mixture of aromatic and aliphatic (non-aromatic) compounds, all of which contribute to the perceived aroma [39]. Essential oils are soluble in alcohol, ether, and fixed oils but insoluble in water. These volatile oils are generally liquid and colorless at room temperature. They have a characteristic odor and have a density less than unity, with the exception of a few cases (cinnamon, sassafras, clove, and vetiver). Moreover, they have a refractive index and a very high optical activity [40].

Essential oils can be composed of only a few to up to more than 100 single substances [41]. Being natural mixtures of a very complex nature, essential oils may consist of about 20–60 components at quite different concentrations, with two or three major components being present at fairly high concentrations (20–70%), compared with other components present in trace amounts [42]. The flavor contribution of single compounds, though, does not strictly depend on their respective concentration but relies on the specific odor threshold that is determined by structure and volatility. Consequently, even minor components deriving from oxidation or degradation reactions may have a strong impact on the flavor if their aroma value is high enough [43]. For the most part, essential oil components can be assigned as lipophilic terpenoids, phenylpropanoids, or short-chain aliphatic hydrocarbon derivatives of low molecular weight, with the first being the most frequent and characteristic constituents. Among these, allylic, mono-, bi-, or tricyclic mono- and sesquiterpenoids of different chemical classes make up a major part of essential oils, such as hydrocarbons, ketones, alcohols, oxides, aldehydes, phenols, or esters [41] (Figure 1). The absence of even one component may change the aroma. In general, monoterpene hydrocarbons are less valuable regarding their contribution to the fragrance of the essential oil than oxygenated compounds, which are highly odoriferous [44]. In addition to their chemical properties, essential oils can be seen to vary greatly in their physical properties in terms of their density, refractive index, and rotatory power. For instance, the essential oils of the different organs (limbo, foliar sheath, and rhizomes) of *Siphonochilus ethiopicus* were reported to significantly differ in those physical properties. These properties contribute to a better characterization of plant essential oils [45].

The terpenes are the largest group of natural fragrances, and their classification is mainly based on the number of isoprene units present in their structure. Depending on the number of C5 units, the terpenes are classified into hemiterpenes (C5), monoterpenes (C10), sesquiterpenes (C15) and diterpenes (C20) [4,46,47]. Monoterpenes are the most abundant in essential oils (about 90%) with a great variety of structures. Geraniol/nerol, linalool, citronellol, citronellal and citral are the most important terpenes and are widely used in the perfume industries [48,49]. Biogenetically, terpenoids and phenylpropanoids have different primary metabolic precursors and are generated through different biosynthetic routes. The pathways involved in terpenoids are the mevalonate and mevalonate-independent (deoxyxylulose phosphate) pathways, whereas phenylpropanoids originate through the shikimate pathway [40].

Many components of essential oils contain one or more asymmetric carbon atoms exhibiting optical activity. These chiral compounds of natural origin (mono- and sesquiterpenes) are usually found in characteristic enantiomeric distributions as they have evolved via enzymatically controlled biosynthetic synthesis. They also have different uses in the food, pharmaceutical as well as fragrance industries. Moreover, the chirality of the odorants exerts an influence on their mode of action, suggesting that, in analogy with other pharmaceuticals, the antipodes may act differently. In addition to variations in their biological, therapeutic, and pharmacological potentials, odor (olfactory) between chiral compounds may differ [50]. For example, the R-(−)- and S-(+)-enantiomers of the monoterpene ketone carvone differ in odor and taste; S-(+)-carvone has an aromatic flavor, while R-(−)-carvone has a characteristic minty flavor. Furthermore, in the case of limonene, the odor of (+)- limonene was rated to be significantly more pleasant and more activating than that of (−)-limonene [50]. As for -(+)- and -(−)-menthol, while both enantiomers have a peppermint odor quality, (−)- menthol has a much stronger cooling effect than (+)-menthol [51].

The chemical composition of essential oils can vary greatly because of physiological (plant organ, ontogenesis), environmental (soil composition, weather conditions), and genetic factors [52]. Other factors that may influence the oils’ chemical composition are geographical variation [53], the plant parameters (e.g., species, cultivated or wild plants), harvest and postharvest/predistillation parameters (e.g., season of harvest, pretreatment of biomass, and storage conditions of source plant materials), and production parameters (e.g., mode of production, distillation parameters, commercial oil, or laboratory-produced), and other parameters (e.g., storage condition and storage time of essential oils, age of essential oil, aging by exposure to oxygen and ultraviolet light, and analytical parameters) [54].

## 5. Classification of Essential Oils Fragrances

Interest in the research of essential oils is mainly championed by the perfume industry. The interest is reflected economically by the readily available unmet demand for fragrances and flavors. This market demand has given rise to the production of synthetic fragrances and flavors. However, the need for natural products has not been overtaken by the supply of synthetic products. In fact, this also has gradually extended to the replacement of purely synthetic products [55].

Essential oils can be classified as top, middle and base “notes” according to their odorous characteristics, diffusion rate in air and volatility [56]. For instance, the top notes are those that are the most volatile and the first perceptible odors [56] that are detected and fade first while providing freshness to the blend. They are responsible for customer first impressions, and therefore, are the selling note of a perfume. They are light scents, lasting 5–10 min or remain for a maximum of 30 min [57]. These include bergamot, juniper, cinnamon, and gardenia.

Middle notes are those essential oils tending to be spicy or floral and give body to blends; their duration time is also brief and can remain up to 1 h. These include ylang-ylang, geranium, lavender, jasmine, and clove.

In contrast, the base notes give a perfume the depth and last the longest. These are the least volatile essential oils that remain longer up to several hours [56,57]. Some essential oils that are used as base notes are myrrh, vanilla, sandalwood, and frankincense [56].

The incorporation of new aromas into perfumes and fragrances and the development of new products are critical to the success of the perfumer and flavor chemist [58]. Indeed, essential oils are important components of perfumes, offering a wide variety of choices for perfume formulations. Perfumes are formulated mostly using alcohol. Eau de types of perfumes are mostly formulated using the essential oils and are usually amber color because of their natural oils color, but they are normally clear [56]. Perfume types can be defined by the amount of essential oil included, e.g., Eau de parfum, Eau de cologne, Eau de toilette, etc. (Table 2).

## 6. Aroma Profiles of Essential Oils and Their Individual Compounds

Essential oils, as well as their isolated compounds, are widely used in cosmetic products as they offer a variety of benefits, including their pleasant aromas. However, the main reason for their usage in cosmetics is their pleasant aromas. Fatty acids, fatty oils and surfactants used in the production-process of cosmetic products often exhibit an unpleasant scent [28]. Essential oils and their terpenoids are one of the most important natural products used by the cosmetic and perfumery industries. In fact, it has been revealed that there is a predominance of essential oils, especially terpenes and terpenoids applied as active ingredients for the products and processes patented in the areas of cosmetics and perfumery [59].

Indeed, essential oils derived from a huge array of aromatic plants have been documented to be used as significant sources of fragrance in perfumes and cosmetics, offering a wide selection of natural scents with unique notes. Hence, in this section, the aroma profiles of essential oils commonly used in perfume and cosmetic industries are presented, along with their principal components, color range, the scientific/common names of plants from which they are derived, and the parts that have been used (Table 3). Besides, the odor description of some individual essential oil fragrances is reported in Table 4.

## 7. Safety of Essential Oils

Mainly due to their perceived safety, the use of essential oils and other botanically derived products has become more popular. However, there have been recent concerns about the standardization and safety of these natural products frequently found in cosmetics and sought as a substitute for synthetic chemicals. For instance, essential oils are challenging to standardize because of the variable growing conditions, genetics, and harvesting of botanicals [102]. In fact, in recent years, several studies have been published with regard to the safety profiles on essential oils [103,104].

Although the vast majority of the most common essential oils have been well tried and tested and safety levels have been determined, there are clearly some essential oils that are more likely to cause adverse reactions than others, and the presence and concentration of a relatively potent allergen is a major factor in allergic contact dermatitis. The presence of adulterants or contaminants may also be a factor. Besides, the oxidation of essential oil constituents can increase risk of causing skin reactions because the oxides and peroxides formed are more reactive. This can be seen with (+)-limonene, δ-3-carene and α-pinene and arise due to the formation of oxidation products, some of which are more sensitizing than the parent compound [103]. For this reason, proper storage of essential oils is required to preserve their effectiveness and decrease the risk of adverse reactions. Accordingly, essential oils should be properly stored in a dark, cool place or refrigerator, in firmly sealed brown bottles [104].

Nevertheless, the fact that an essential oil constituent is capable of being oxidized does not automatically imply that the use of essential oil containing it presents a significant allergenic risk [103]. On the other hand, many essential oils that are considered to be non-toxic can have a toxic effect on some people; this can be influenced by preceding sensitization to a particular essential oil, a group of essential oils having similar components or some adulterants in an essential oil [105].

Photosensitization may also occur when a phototoxin in the essential oil is applied to the skin in the presence of sunlight or ultraviolet A (UVA) light. For instance, furanocoumarins are phototoxins that can be found mainly in expressed citrus peel oils, although they may be present in angelica root (*Angelica archangelica* L.), rue (*Ruta graveolens* L.), parsley leaf (*Petroselinum crispum* (Mill.) Fuss), and marigold (*Tagetes minuta* L.) essential oils. The most common furanocoumarins are psoralen and bergapten. Inflammatory skin reactions occurring when an essential oil containing these phototoxic compounds applied to the skin is exposed to UVA light can vary from pigmentation, blistering to severe full-thickness burns [106]. While some essential oils are known as dermal irritants, their severity may depend on their concentration. These should not be used on any inflammatory or allergic skin condition and should always be appropriately diluted in vegetable oils. Moreover, since some essential oils can make the skin more sensitive when exposed to the sun and UV rays like those encountered in a tanning bed, exposure to the sun or tanning salon should be avoided for at least 24 h after application of photosensitizing essential oils [107].

Twenty-six possible allergenic fragrances have been defined [108] (Table 5), of which 18 can be found as ingredients of essential oils, and for this reason, they must be declared on the packaging or in the information leaflet if the concentration of these allergenic fragrances is higher than the permissible concentration of 0.01% in shower gels and baths (rinse-off products) and higher than 0.001% in body oils, massage oils and creams (leave-on products) [28]. For example, methyl eugenol should not be intentionally added as a cosmetic ingredient. However, when fragrance compounds containing methyl eugenol present naturally in essential oils are used as components in cosmetic products, the highest concentration of methyl eugenol in the finished products must not go beyond 0.01% in fine fragrance, 0.004% in eau de toilette, 0.002% in a fragrance cream, 0.0002% in other leave-on products and in oral hygiene products, and 0.001% in rinse-off products [109].

Regulatory authorities such as the Food and Drug Administration (FDA) do not require approval for a fragrance that may contain essential oil since fragrances are regarded as cosmetic ingredients. Compared to food products and drugs, the regulation process of the FDA for essential oils that are used as fragrance ingredients is not that rigorous. Consequently, cosmetic manufacturers have an important responsibility to provide high-quality products. The FDA always requires a list of ingredients to ensure product quality; though, it will not oblige the manufacturer to divulge cosmetic secrets because the FDA does not have the authority to require allergen labeling for cosmetics. This situation may be disadvantageous to consumers, particularly to those with less familiarity with essential oils [110].

Furthermore, the International Organization for Standardization (ISO) codifies the composition of each essential oil with an individual identification (e.g., ISO1342:1988 for *Rosmarinus officinalis* L. and ISO11043:1998 for *Ocimum basilicum* L.). In ISO/Technical committee 54, ISO has also standardized many analytical methods for controlling the quality of these specifications along with requirements for labeling, transport, and marking. ISO/TC 54 emphasizes global trade in essential oils, thus focusing on improving the quality of the essential oil market and protecting the health of consumers who buy products containing essential oils; to increase the safety level of these oils [110].

Additionally, the International Fragrance Association (IFRA) defines which essential oils and which components represent a potential allergy risk and determines their maximum concentration to produce safe cosmetic products [28]. IFRA also issues recommendations for the safe use of fragrance ingredients, which are published in the IFRA Code of Practice and its guidelines [111].

Essential oil safety is monitored in a variety of different ways, all of which have been geared to the perfumery, cosmetics and food industries. The synthesis of new aroma chemicals and their widespread usage in “natural essential oils” together with many diluents, has brought about many problems, the worst being dermatologic sensitivity [112].

Because of the complexity of essential oils, however, it is important to test specific oils, given that fragrance mixes alone are not sufficient for the elimination of essential oil contact allergy [54,113]. It is often difficult to accurately determine the responsible allergen due to the complexities of the chemical compositions and the co-reactivity with other fragrances. While the screening series includes several popular essential oils, it is important to test patients with their own products, as their composition may have been altered by the aging process. Testing can be achieved safely for most essential oils by the formulation of 2% to 5% oil concentrations in petrolatum [54]. Patch testing can do much to detect and avoid skin reactions. The amount of essential oil used is usually measured in drops or percentage. On average, there are 20 drops of essential oil in 1 mL. Patch tests can be used to test for irritation and are suggested for all potential-risk patients. For this purpose, the essential oil is diluted at double the concentration desired to be used and applied on an adhesive bandage and placed on the forearm. If irritation is bound to happen, it will do so quickly [106].

## 8. Authenticity of Essential Oils

One of the chief problems with large-scale essential oils’ production is the relatively small yield per unit mass of raw materials that seemingly result in the generally high prices that the oils fetch on the international market [55]. This, coupled with the increasing demand for essential oils, are one of the obvious reasons for essential oil adulteration. Adulteration could feasibly increase toxicity, mainly in the area of skin reactions. The co-presence of both adulterants and contaminants is also a matter of concern [103]. Authentication is therefore of crucial significance for both consumers and chemical companies [114]. Hence, this part of the review will cover some of the known cases of adulterations in essential oils and some existing analytical techniques adopted for their detection.

In the course of time, a great number of techniques have been developed and applied for essential oils’ analyses. While part of them has been replaced by either easier or easier-to-handle techniques, other methods have maintained their importance and have been continuously improved [115].

Essential oils have long been analyzed by measuring their physicochemical constants. These values indicate the quality of the oils and are critical for the detection of adulteration. The physical constants include the specific gravity, refractive index and optical rotation. Solubility is expressed in terms of solubility in ethyl alcohol and has a practical significance. On the other hand, the chemical constants used to assess essential oils are acid value, ester value after acetylation, as well as alcohol, aldehyde, and ketone contents, etc. These parameters demonstrate the characteristics of essential oils and are greatly helpful in evaluating their quality (Figure 2) [3].

Gas chromatography (GC) and mass spectrometry (MS) are the most common and important tools used to analyze the constituents of essential oils and can add further improvements to essential oil analysis where broad categories of monoterpenes and varying isomers strongly resembling each other in terms of their chemical structures can be individually detected [116]. In fact, structural isomers, as well as *cis*–*trans* isomers, are generally separated in GC with standard columns and hence, is the method of choice for the analysis of essential oils [117]. In addition, optically active compounds are detectable by chiral chromatography [118]. GC is ideally suited to volatile compounds and has revolutionized the detection of minor chemical constituents, especially when used in conjunction with MS and nuclear magnetic resonance (NMR) spectroscopy. MS looks at the fragmentation patterns of compounds under ionizing conditions, and this information is used to deduce their structures, while NMR elucidates the structures of molecules by examining the environment of specific atoms such as hydrogen, by looking at their characteristic nuclear spins. The sensitivity of analytical techniques for organic compounds has considerably increased over recent years to the point whereby even trace constituents, including pollutants such as pesticides, can be detected as well [103].

A variety of chromatographic columns and detectors are used in GC that has indeed helped to expand its applicability to various analytical problems, including essential oil analysis. Different types of detectors (selective or not) can be used to generate signals. The flame ionization detector (FID) is the most popular detector for GC for its reliability and its sensitivity in the detection of organic vapors and is non-selective compared to many specific detectors. FID detectors are mass rather than concentration sensitive and can detect most carbon-containing substances, which makes them very convenient for the analysis of the majority of essential oils [119].

There is no simple generalization about the key odorants in essential oils; the odor of some essential oils is due to a large amount of a single compound, although trace amounts could determine the odor of others, and in general, the true odor is the manifestation of a complex mixture of compounds [120]. Gas-chromatography-olfactometry (GC-O) is a well-known standard technique that enables the assessment of odor-active components in complex mixtures, based on the correlation between the chromatographic peaks of the eluted substances perceived simultaneously by two detectors, one of them being the human olfactory system. Over the years, the GC-O has been frequently employed in essential oil analysis [100,121,122]. Results of GC-O analysis can give information about the presence or absence of odor in a compound, measure the duration of odor activity, describe the quality of the odor perceived, and quantify the intensity of specific odorant which may have an ultimate purpose in the application of such compounds in the flavor and fragrance industries. GC-O is also used to locate the odor-active regions in the chromatogram and to generate an odor profile for the whole essential oil sample [123].

High-Performance liquid chromatography-Ultraviolet (HPLC-UV) may represent a supplementary or even alternative method for analysis of volatile oils because of their sensitivity, versatility, and selectivity [124]. Usually, HPLC is the method of choice in the analysis of less volatile constituents of essential oils [125]. The use of HPLC-MS provides valuable information about the content and nature of constituents of natural complex matrices, such as essential oils [126].

Indeed, several reported cases of essential oil adulteration have been documented, such as the addition of a non-volatile ingredient, cheap synthetic compounds, volatile or essential oil from other natural sources, as well as vegetable oils so as to increase their weight. Other than that, the total or partial substitution of part of the original plant from which the essential oil is obtained by other plants has also been reported [52,127]. All these adulteration methods can degrade the quality of the essential oils and, by adding one or more synthetic compounds, adulteration can lead to safety issues or non-compliance with the natural labels. Consequently, authentication is an important topic for consumer protection and the quality of essential oil production [128]. Adulteration of essential oils can also have an effect on the regulatory aspect, as an essential oil may no longer conform to specifications of standardization. Most of the time, adulterants are added at a low level (5–8%) to avoid detection by common analytical methods [129].

## 9. Conclusions

Essential oils have seen a revival in popularity in the last few years. They are widely used in the cosmetic industries as a fragrance and active components. Moreover, their ability to impart a wide range of unique and pleasant aromas in cosmetic products and at the same time acting as bioactive agents (anti-aging, antimicrobial, sun protection, and whitening) make them prized and highly valued ingredients in cosmetics and cosmeceutical products.

Moreover, the ‘’back to nature” trend has enormously amplified the use of botanical extracts and oils at the cost of artificial and synthetic derivatives, thought to be hazardous to human health. Furthermore, unlike artificial fragrances, which can mimic some herbal fragrances, essential oils are nonetheless more in demand given the increasing awareness of their health benefits supported by scientific findings, thus making them more tempting and attractive to consumers. On that note, the future of the essential oils industry seems promising, with lucrative avenues in the cosmetic and perfume industries.

However, although essential oils are generally regarded as safe, as complex mixtures of compounds, some of which known to be allergens and skin sensitizing agents, need to be indicated on cosmetics labels, especially for consumers having sensitive, allergic-prone skin or existing skin disorders and could opt for patch testing before using products containing them.

## Figures and Tables

**Figure 1 molecules-26-00666-f001:**
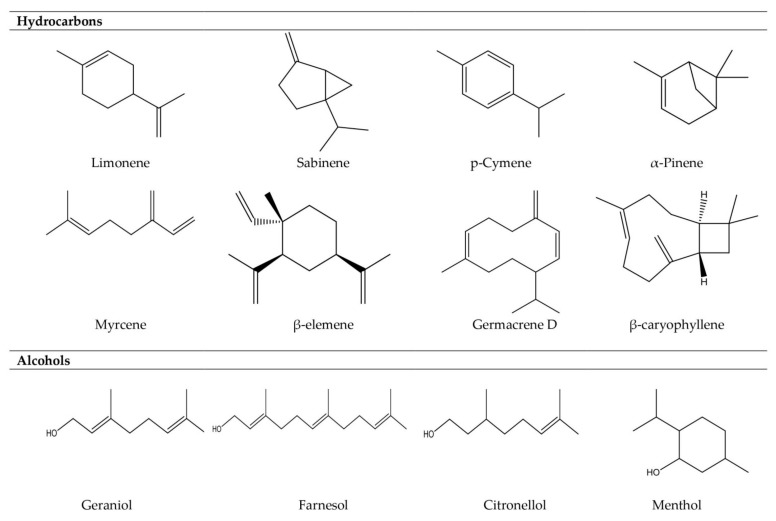
Examples from major classes of compounds in essential oils.

**Figure 2 molecules-26-00666-f002:**
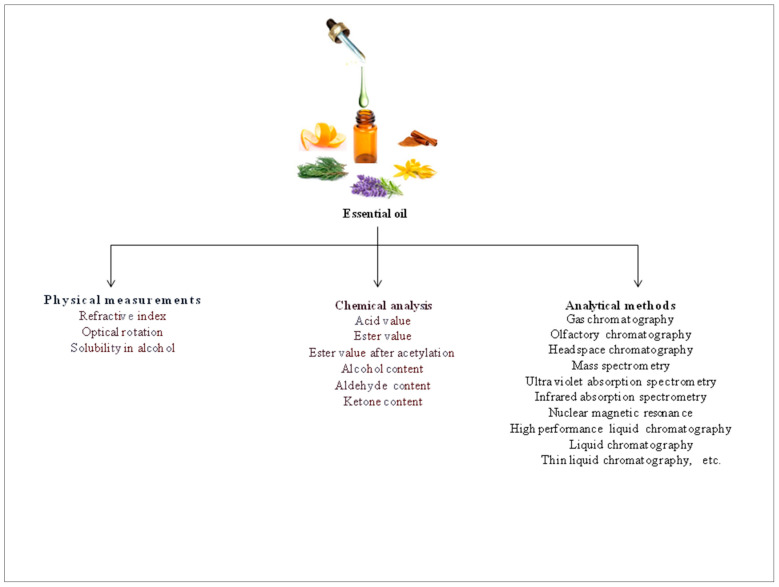
Analysis of essential oils.

**Table 1 molecules-26-00666-t001:** Main sources of essential oils in plant families [32].

Type of Vascular Plants	Plant Families	Examples of Essential Oil-Bearing Plants
1. Gymnosperms	Cupressaceae	Cedar leaf, cedarwood, juniper
	Pinaceae	Fir and pine
2. Angiosperms		
Monocots	Acoraceae	Calamus
	Poaceae	Vetiver and aromatic grass
	Zingiberaceae	Ginger and cardamom
Dicots	Apiaceae	Coriander and fennel
	Asteraceae	Tarragon, chamomile and wormwood
	Geraniaceae	Geranium
	Illiciaceae	Star anise
	Lamiaceae	Lavender, patchouli, mint, oregano
	Lauraceae	Litsea, cinnamon, camphor, sassafras
	Myristicaceae	Mace and nutmeg
	Myrtaceae	Allspice, myrtle and clove
	Oleaceae	Jasmine
	Rosaceae	Rose
	Santalaceae	Sandalwood

**Table 2 molecules-26-00666-t002:** Varying percentage of essential oils and alcohol in different perfume types [56].

Type of Perfume	Fragrance/Essential Oil	Alcohol
Eau de parfum	8–15%	80–90%
Splash colognes	1–3%	80%
Eau de cologne	3–5%	70%
Eau de toilette	4–8%	80–90%

**Table 3 molecules-26-00666-t003:** Aroma profile of essential oils used as fragrances in cosmetics and/or perfumes.

Plant Species from Which Essential Oils are Derived	Common Name	Part(s) Used	Principal Components^a^	Color Range of Oils	Aroma Description	References
1. *Amyris balsamifera* L.	Balsam torchwood	Wood	Elemol, 10,γ-*epi*-eudesmol, γ-eudesmol, valerianol, α-eudesmol, 7-α-*epi*-eudesmol, β-eudesmol, drimenol	Pale yellow to amber yellow	Characteristic, woody	[60]
2. *Angelica archangelica*	Angelica	Roots and rhizomes	Phellandrene, pinene, limonene, linalool and borneol; rich in coumarins including osthol, Angelicin, bergapten and imperatorin	Colorless or pale yellow	Rich herbaceous-earthy body note	[61]
		Seed		Colorless	Fresher, spicy top note	
3. *Aniba rosaeodora* Ducke	Rosewood	Wood/leaves	α-Pinene, β-pinene, limonene, 1,8-cineole, trans-linalool oxide (furanoid), cis-linalool oxide (furanoid), linalool, trans-linalool oxide (pyranoid), α-terpineol, α-copaene, β-selinene, α-selinene, spathulenol, caryophyllene oxide, benzyl benzoate	Pale-yellow	Floral, sweet, woody, and citric	[62,63]
4. *Boswellia sacra* Flueck.	Frankincense	Gum	α-Pinene (2.0–64.7%), α-thujene (0.3–52.4%), β-pinene (0.3–13.1%), myrcene (1.1–22.4%), sabinene (0.5–7.0%), limonene (1.3–20.4%), *p*-cymene (2.7–16.9%) and β-caryophyllene (0.1–10.5%)	Clear or yellow	Balsamic, camphor-like, spicy, woody, slightly lemony	[64,65]
5. *Cananga odorata* Hook. F. and Thoms)	Ylang ylang	Flowers	Prenyl acetate, *p*-cresyl methyl ether, methyl benzoate, linalool, benzyl acetate, geraniol, geranyl acetate, (*E*)-cinnamyl acetate, β-caryophyllene, germacrene-D, (*E,E*)-α-farnesene, (*E,E*)-farnesol, benzyl benzoate, (*E,E*)-farnesyl acetate and benzyl salicylate	Pale yellow to dark yellow	Characteristic, floral, recalling jasmine	ISO3063:2004(E)
6. *Carum carvi* L.	Caraway	Ripe fruit	Myrcene, limonene, *cis*-dihyrocarvone, *trans*-carveol, *cis*-carveol, carvone	Colorless to pale yellow	Fresh, herbaceous, spicy	[66]
7. *Cedrus atlantica* (Endl.) Manetti ex Carrière	Cedarwood	Wood/sawdust	Himachalene, α-himachalene, β-himachalene, *cis*-bisabolene, himachalol, allo-himachalol, α-atlantone, ƴ-atlantone, himachalene oxide	Yellowish to orange-yellow or deep amber-colored	Camphoraceous–cresylic top note with a sweet, tenacious woody undertone, reminiscent of cassie and mimosa	[67]
8. *Cinnamomum verum* J. Presl	Cinnamon	Bark	(*E*)-Cinnamaldehyde, eugenol, β-caryophyllene, β-phellandrene, linalool	Reddish-brown	Characteristically spicy burning	[54,64]
9. *Citrus aurantiifolia* (Christm.) Swingle	Lime	Peel	Limonene, β-pinene, γ-terpinene, citral	Colorless to greenish-yellow	Mild citrus, floral	[10,68]
10. *Citrus aurantium* var. *amara* L.	Neroli/bitter orange	Flowers	Linalool, β-pinene, α-terpineol, limonene, sabinene, nerol, nerolidol, linalyl acetate and α-pinene	Pale yellow to coffee brown	Sweet, fresh and floral odor	[28,69]
11. *Citrus bergamia* Risso	Bergamot	Rind of fruit	Linalyl acetate, limonene, α-terpinene, β-pinene, γ-terpinene, geraniol, linalool, lactones, bergaptene, β-bisabolene	Light green-yellow	Light, delicate citrus-like scent with slightly floral/sweet/balsamic character (top note)	[11]
12. *Citrus limon* (L.) Osbeck	Lemon	Peel	Limonene, β-pinene, γ-terpinene, sabinene, α-pinene, geranial, neral, α-thujene, β-bisabolene, terpinolene, trans-α-bergamotene, α-terpineol, α-terpinene, neryl acetate, linalool, p-cymene, citronellal, trans-caryophyllene, terpineol-4, nerol, camphene, nonanal, geraniol, octyl aldehyde, α-phellandrene, and cis-sabinene-hydrate	Colorless to pale yellow	Fresh lemon peel	[10,70]
13. *Citrus reticulata* Blanco	Tangerine	Peel	Limonene, γ-terpinene, terpinolene, myrcene, α-pinene, *p*-cymene, α-thujene	Dark orange to reddish-orange or brownish-orange	Orange-like	[10,54]
14. *Citrus sinensis* (L.) Osbeck	Sweet orange	Peel	Limonene, α-pinene, β-pinene, sabinene, myrcene, geranial, n-octanal, n-decanal, n-nonanal, neral, β-sinensal, valencene, linalool	Yellow to reddish-yellow	Characteristic, orange peel odor	[71]
15. *Citrus paradisi* Macfad.	Grapefruit	Peel of fruit	Limonene, citronellal, citral, sinensal, geranyl acetate, paradisol, geraniol, ketones, lactones, coumarins such as auraptene, limettin and sesquiterpenes such as cadinene	Yellow or greenish	Fresh, sharp, citrus aroma (top note)	[11]
16. *Commiphora myrrha* (Nees) Engl.	Myrrh	Gum	Furanoeudesma-1,3-diene, curzerene, β-elemene, lindestrene, furanodiene	Yellow to reddish-brown	Pleasant balsamic, camphor-like, musty, incense-like	[54,64]
17. *Coriandrum sativum* L.	Coriander	Fruit	Borneol, 2E-decenal, camphor, dodecanal, n-decanal, neryl acetate, geraniol, linalool	Colorless to pale yellow	Floral, balsamic undertone and peppery-woody suave top-note	[72,73]
18. *Cupressus sempervivens* L.	Cypress	Leaves	α-Pinene, Δ-carene, limonene, sesquiterpene, α-terpinene, sabinene, carvone	Yellowish	Pleasingly smoky, woody smell, and amber-like	[74,75]
19. *Cymbopogon citratus* (DC.) Stapf	Lemongrass	Grass leaf	Citral, myrcene, dipentene, linalool, geraniol, nerol, citronellol, and farnesol, sesquiterpenes, methyl heptenone, esters, acids and others	Yellow or pale sherry-colored	Intense sweet lemony, reminiscent of lemon drops (top note)	[11]
20. *Cymbopogon martini* (Roxb.) W. Watson	Palmarosa	Leaves	Myrcene, linalool, β-caryophyllene, α-terpineol, geranyl acetate, geranyl isobutyrate, geraniol, (*E,Z*) farnesol, geranyl hexanoate	Yellow	Fresh rose-like	[64,76]
21. *Daucus carota* L.	Carrot seed	Seeds	β-Bisabolene, elemicin, geranyl acetate, and sabinene	Yellowish-brown color	Soft, sweet earthy grounding aroma	[77]
22. *Elettaria cardamomum* (L.) Maton	Cardamom	Fruits	α-Pinene, sabinene, myrcene, limonene, 1,8-cineole, linalool, linalyl acetate, terpinen-4-ol, α-terpineol, terpinyl acetate, *trans*-nerolidol	Almost colorless to pale yellow	Characteristic, spicy, cineolic	[78]
23. *Eucalyptus globulus* Labill.	Eucalyptus	Leaves	1,8-Cineole, α-pinene, limonene, aromadendrene, p-cymene, globulol	Yellow to red	Fresh balsamic camphor-like	[54,64]
24. *Feniculum vulgare* Mill.	Fennel	Seeds	Phenols, trans-anethole, methyl chavicol, α-pinene, α-thujene, γ-terpinene, limonene, myrcene, phellandrene, fenchone, 1,8-cineole, fenchol, acids, lactones and coumarins	Yellowish	Strong, anisey, camphoric	[11,64]
25. *Helichrysum italicum* (Roth) G. Don	Immortelle	Aerial parts	Neryl acetate, g-curcumene, neryl propionate and ar-curcumene	Pale yellow to red	Strong, honey-like aroma	[28,79]
26. *Hyssopus officinalis* L.	Hyssop	Aerial parts	Isopinocamphone, pinocamphone, β-pinene	Light-yellow	Herbaceous, camphor-like odors with warm and spicy undertones	[80]
27. *Illicium verum* Hook.f.	Star anise	Fruit	Trans-anethole (80–90%)	Pale yellow	Warm, spicy, extremely sweet, licorice-like scent	[61]
28. *Jasminum officinale* L.	Jasmine	Flowers	Benzyl acetate, linalyl acetate, benzyl benzoate, methyl jasmonate, methyl anthranilate, linalool, nerol, geraniol, benzyl alcohol, farnesol, terpineol, phytols, eugenol, *cis*-jasmone; acids, aldehydes and others	Dark, orange-brown	Highly intense, rich, sweet, floral odor (base note)	[11]
29. *Juniper communis* L.	Juniper	Ripe berries	α-Pinene (20–50%), sabinene (<20%), β-pinene (1–12%), β-myrcene 1–35%, α-phellandrene (<1%), limonene 2–12%, terpinen-4-ol (0.5–10%), bornyl acetate (<2%), and β-caryophyllene (< 7%)	Colorless or pale yellow-green	Fresh terebinth or turpentine-like/conifer-like aroma	[74,81]
30. *Laurus nobilis* L.	Laurel	Leaves	1,8-Cineole, sabinene, α-terpinyl acetate, linalool, eugenol, methyl eugenol, α-pinene	Yellow	Aromatic, spicy	[82]
31. *Lavandula angustifolia* Mill.	Lavender	Flowering tops	Linalyl acetate (25–47%), linalool (max. 45%), terpinen-4-ol (max. 8%), camphor (max. 1.5%), limonene (max. 1%) and 1,8-cineole (max. 3%)	Colorless to pale yellow	Sweet floral aroma	[28]
32. *Litsea cubeba* (Lour.) Pers.	Litsea	Fruits	Citral (neral and geranial) (78.7–87.4%), d-limonene (0.7–5.3%)	Pale yellow	An intense lemon-like, spicy aroma	[83]
33. *Matricaria chamomilla* L.	German chamomile	Flowers and flower heads	(*E*)-β-Farnesene (4.9–8.1%), terpene alcohol (farnesol), chamazulene (2.3–10.9%), α-bisabolol (4.8–11.3%), and α-bisabolol oxides A (25.5–28.7%) and α-bisabolol oxides B (12.2–30.9%)	Blue	Warm, fruity scent Sweet herbaceous odor (middle note)	[84]
34. *Melaleuca alternifolia* (Maiden and Betche) Cheel	Tea tree	Leaves	Terpinen-4-ol (30–48%), γ-terpinene (10–28%), 1,8-cineole (traces-15%), α-terpinene (5–13%), α-terpineol (1.5–8%), *p*-cymene (0.5–8%), α-pinene (1–6%), terpinolene (1.5–5%), sabinene (traces-3.5%), aromadendrene (traces-3%), δ-cadinene (traces-3%), viridiflorene (traces-3%), limonene (0.5–1.5%), globulol (traces-1%), viridiflorol (traces-1%) [ISO 4730 (2004)]	Colorless to pale yellow	Intensive aromatic fresh camphoraceous odor	[28]
35. *Melaleuca leucadendra* (L.) L.	Cajeput	Fresh leaves and twigs	1,8-Cineole (14–65%), terpenes (45%), alcohols (5%), esters	Pale yellow-green	Camphorous, highly penetrating odor with a slightly fruity note (top note)	[11]
36. *Mentha* x *piperita* L.	Peppermint	Aerial parts	Menthol, menthone, isomenthone, menthofuran, neomenthol, pulegone, menthyl acetate, 3-octanal, 1,8-cineole, limonene, *trans*-sabinene hydrate, β-caryophyllene	Almost colorless to pale greenish yellow	Characteristic of mint, sweet, menthol-like	[85]
37. *Myristica fragrans* Houtt.	Nutmeg	Seeds	Myristicin, α-pinene, sabinene, limonene, elemicin, eugenol, safrol and β-pinene	Pale yellow to nearly colorless	Spicy, sweet, woody	[86]
38. *Ocimum basilicum* L.	Basil	Flowering herb	Linalool, citronellol, geraniol, terpinen-4-ol, α-terpineol, methyl chavicol, eugenol, ethyl eugenol, limonene, camphene, α-pinene, β-pinene, γ-terpinene, p-cymene, cis-ocimene, 1,8-cineole, linalyl acetate, fenchyl acetate, methyl cinnamate, β-caryophyllene	Colorless or pale yellow	Light sweet, spicy odor reminiscent of anise or clove (top note)	[11]
39. *Pelargonium graveolens* L’Hér.	Geranium	Aerial parts	Citronellol, geraniol, linalool, citronellyl formate, geranyl formate, citral, guaiazulene, β-caryophyllene, *cis*-rose oxide, phellandrene, limonene, α-pinene, menthone	Pale or olive green	Sweet rose-like odor with a hint of mint or “greenness” (middle note)	[11]
40. *Pimenta dioica* (L.) Merr.	Allspice	Leaf	Mainly eugenol, less in the fruit (60–80%) than in the leaves (up to 96%), also methyl eugenol, cineole, phellandrene and caryophyllene, among others	Yellowish-red or brownish liquid	Powerful sweet, spicy scent, similar to cloves	[61]
		Berry		Pale yellow liquid	Sweet, warm balsamic, spicy body note (middle note) and fresh, clean top note	
41. *Pimpinella anisum* L.	Aniseed	Seeds	Trans-anethole (75–90%)	Colorless to pale yellow	Warm, spicy-sweet characteristic scent	[61]
42. *Pinus massoniana* Lamb.	Turpentine	Gum resin	α-Pinene, camphene, β-pinene, δ-3-carene, myrcene, limonene, p-cymene, longifolene, β-caryophyllene, caryophyllene oxide	Colorless	Characteristic of gum turpentine	[87]
43. *Piper nigrum* L.	Black pepper	Unripe berries (peppercorns)	Sesquiterpenes (20–30%) and monoterpenes (70–80%) such as bisabolene, β-caryophyllene, thujene, pinene, camphene, sabinene, terpinene, myrcene, limonene, phellandrene, small amounts of ketone, phenols, alcohols and others	Light to pale olive	Dry, woody, warm, spicy, oriental (base note)	[11]
44. *Pogostemon cablin* (Blanco) Benth.	Patchouli	Leaves	α-Pinene (0.01–0.3%), β-pinene (0.02–1%), Limonene (0.01–0.3%), δ-elemene (0.01–1.9%), β-patchoulene (0.03–12%), β-elemene (0.18–1.9%), cycloseychellene (0.02–0.8%), (*E*)-β-caryophyllene (0.75–6.8%), α-guaiene (2.9–23%), seychellene (2.3–13%), α-humulene (0.05–2%), α-patchoulene (1.2-13%), pogostone (0.1–27.7%), patchoulol (11–72%), pogostol (0.2–6.2%), α-bulnesene (2.9–23%), aciphyllene (0.7–4.2%), norpatchoulenol (0.11–4.0%), caryophyllene oxide (0.0–4.6%), germacrene D (0.0–0.2%)	Yellow to reddish-brown	Earthy, woody and camphoraceous	[88]
45. *Rosa* *damascena* Mill.	Rose	Flowers	Citronellol, geraniol, nerol, nonadecane, heneicosane	Yellow to yellow-green	Sweet, floral, rosaceous	[89,90,91]
46. *Rosmarinus officinalis* L.	Rosemary	Leaves	Eucalyptol and α-pinene, camphor, bornyl acetate, camphene, β-pinene, β-myrcene, limonene and borneol	Colorless to pale yellow	Strong, warm, woody, balsamic aroma	[28,92]
47. *Salvia sclarea* L.	Clary sage	Inflorescences	Linalool, linalyl acetate	Yellow	Characteristic herbaceous odor	[93,94]
48. *Santalum album* L.	Sandalwood	Heartwood	90% Santalol content (cis-α-santalol and *cis-β-*santalol) (ISO 3518:2002)	Yellow to light brown	Sweet woody	[95]
49. *Syzygium aromaticum* (L.) Merrill et. Perry	Clove	Bud	Eugenol (70–95%), eugenol acetate (up to 20%) and β-caryophyllene (12–17%)	Colorless or pale yellow	Clove aroma	[96]
50. *Vetiveria zizanioides* (Linn.) Nash	Vetiver	Roots	α-Vetivone (8.4–13.3%), khusimol (0.6–8.9%), β-vetivone (2.2–3.7%), khusian-2-ol (1.8–2.3%) and khusimone (1.2–2.3%)	Pale-yellow to dark brown, olive or amber	Deep, smoky, earthy, and woody with a sweet persistent undertone	[30,97,98]

^a^ Essential oils’ chemical composition is subject to variation according to changes in extrinsic and extrinsic factors [40].

**Table 4 molecules-26-00666-t004:** Odor characteristics of some individual compounds of essential oils.

Essential Oil Compounds	Odor Description	Reference
Anethole	Pleasant odor of anise oil	[28]
Bisabolol	Sweet floral odor	[28]
Bornyl acetate	Woody, camphor, mentholic, spicy	[99]
Carvone	Minty herbaceous	[100]
Citral	Lemony	[28]
Citronellol	Strong floral, rose, sweet like	[101]
Cuminal	Spicy harsh	[101]
Decanal	Sweet, aldehydic, orange, waxy, citrus rind	[99]
Farnesol	Flowery, weak-citrus odor	[28]
Geraniol	Fresh, sweet, rose-like	[101]
Geranyl acetate	Pleasant, floral rose, herbal	[101]
Germacrene D	Woody, spice	[99]
Limonene	Strong odor of orange	[28]
Linalool	Floral, grassy, pleasant, citrus	[101]
Linalyl acetate	Floral, sweet citrus	[100]
Menthol	Sweet minty, cooling and fresh scent	[28]
Myrcene	Pleasant floral	[101]
Myristicin	Spice, warm, balsam, woody	[99]
Neral	Citric, green	[100]
*p*-Cymene	Fresh, citrus, terpene, woody, spice	[99]
Sabinene	Woody, terpene, citrus, pine, spice	[99]
Terpineol	Sweet, lilac odor	[101]
Terpinolene	Fresh, woody, sweet, pine, citrus	[99]
α-Pinene	Fresh, camphor, sweet, pine, earthy, woody	[99]
β-Phellandrene	Mint, turpentine	[99]
β-Pinene	Woody, turpentine	[101]
γ-Terpinene	Woody, terpene, lemon, lime, tropical, herbal	[99]

**Table 5 molecules-26-00666-t005:** List of 26 possible allergenic fragrances [28].

Amylcinnamal	Geraniol
Amylcinnamyl alcohol	Farnesol
Anisyl alcohol	Hexyl cinnamaldehyde
Benzyl alcohol	Hydroxy-citronellal
Benzyl benzoate	Hydroxy-methylpentylcyclohexenecarboxaldehyde
Benzyl cinnamate	Isoeugenol
Benzyl salicylate	D-Limonene
Cinnamyl alcohol	Linalool
Cinnamal	Methyl heptin carbonate
Citral	3-Methyl-4-(2,6,6-tri-methyl-2-cyclohexen-1-yl)-3-buten-2-one
Citronellol	Oakmoss and treemoss extract
Coumarin	treemoss extract
Eugenol	2-(4-tert-Butylbenzyl) propionaldehyde

## Data Availability

The data presented in this study are available in this article.

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
