# Peer review of "Essential Oils as Natural Sources of Fragrance Compounds for Cosmetics and Cosmeceuticals"

_molecules, 2021, doi:10.3390/molecules26030666_

Round 1

Reviewer 1 Report

This review on eseential oils as sources for fragrance compounds in cosmetics by Sharmeen et al. is informative and with appropriate revisions  it could be published.

Specifically, I suggest the authors change the Abstract to reflect what is in the main article. Also, the Introduction should describe how their review focuses on a different aspect than other recent reviews on essential oils.

The review needs to be better organized and the English language improved. The authors need to proofread the manuscript more before submitting to the journal. There were several errors that should have been avoided.

More specific changes are:

The title should be revised - remove the word "products"

line 49-50 - rewrite. Not clear what the "old scent" means

line 56-59 too long

line 63 - define what expression is

line 71-change "of" to "a"

line 75 - remove "amongst others"

line 81- change "has also been" to "is"

line 87 change "in" to " to"

line 89 change "given that they are" to "that are"

line 90 remove "than their synthetic counterparts" also, Start new sentence with "Synthetic fragrances..."

line 93-94 modify to read "and known carcinogens such as benzene derivatives " 

line 103 change "the distinguishable from others as"  to "them unique as"

line 112 change to " putting biodiversity at risk[26].

Table 1 under Lamiaceae remove or clarify which other herbs " including other herbs"

line 133 remove "constantly"

line136 remove "the health industry in areas of "

line 189 separate odor and

line204 change "in" to "of"

line 208 - give name of limbo plant as it is not common

line 215 remove "In the terpenes"

line 250 change "in the research in" to " on research of"

line 254 change "extend" to "extended"

line 259 change to "first while providing"

line 261 change "remains" to "remain"

line 264 change "also less" to "brief"

line 266 change to "In contrast, the base notes gives a perfume its"

line 267 change to " remain longer up to"

line 268 remove " the"

line 281 change to "benfits, including their pleasant aromas."

line 292 change "among" to "of"

in table 2 number 29 Juniper.... change to "pale yellow"

table 3 linaool AND linalyl acetate change to "citrus"

line 297 should be number 6 (not 7)

line 299  remove "progressively"

line 301 remove "to"

line 305 change "commonest" to "most common"

line 337 make a table listing the 26 possible allergens

line 348 change "does" to "do"

line352 should be "manufacturers have" and change "make sure of the" to "have"

line 353 add "ensure product quality"

line 354 change "pass" to "divulge"

line 355 add "be advantageous"

line 362 remove "methods and"

line 363-366 is it necessary to define what the ISO does? needs to be rewritten (and shortened) if so.

line 368 remove "of them" change "the maximum concentration of them" to " their maximum concentration"

line 372 change "has been " to "is"

line 373 remove continuous

line 375 add "dermatologic sensitivity"

line 377 change "With" to "Because of " and change to " to test specific"

line 382 change "against" to "with"

line 388 change " is diluted to double the concentration to be used regularly" to "is diluted to double the concentration," 

line 391 should be 7.

line 392-428 this section needs to be rewritten - especially the grammar and spelling needs to be corrected

line 424 a description of NMR is not needed for the audience that reads the journal Molecules

line 429-430 - need to rewrite as it is not clear

line 458-461 needs to be rewritten for English grammar.

line 458 remove "over the time"

line 462 change "natural grade" to ""natural" labels"

line 467 number is incorrect for the section

line 468-469 remove "Indeed"  and remove "and do not appear to be passing so soon"

line 475 change "apprehended as being" to "thought to be"

line 476 change "even though are able to" to "can"

line 484 change "watched out" to "indicated" and change "by" to "for"

Author Response

Comments from reviewer #1: Revisions in the manuscript are highlighted in yellow

This review on essential oils as sources for fragrance compounds in cosmetics by Sharmeen et al. is informative and with appropriate revisions it could be published.

Thank you for your valuable comments and granting us the opportunity to share these data to the scientific community.

  1. Specifically, I suggest the authors change the Abstract to reflect what is in the main article. Also, the Introduction should describe how their review focuses on a different aspect than other recent reviews on essential oils.

The abstract has been amended as per suggestions.

  1. The review needs to be better organized and the English language improved. The authors need to proofread the manuscript more before submitting to the journal. There were several errors that should have been avoided.

Review has been checked for errors and overall English has been improved.

More specific changes are:

  1. The title should be revised - remove the word "products"

Title has been revised.

  1. Line 49-50 - rewrite. Not clear what the "old scent" means

The sentence has been removed.

  1. Line 56-59 too long

The sentence has been rephrased.

  1. Line 63 - define what expression is

Definition has been given in brackets.

  1. Line 71-change "of" to "a"

Change has been made.

  1. Line 75 - remove "amongst others"

Words have been removed.

  1. Line 81- change "has also been" to "is"

Change has been made.

  1. Line 87 change "in" to "to"

Change has been made.

  1. Line 89 change "given that they are" to "that are"

Change has been made.

  1. Line 90 remove "than their synthetic counterparts" also, Start new sentence with "Synthetic fragrances..."

Sentence has been rephrased as suggested.

  1. Line 93-94 modify to read "and known carcinogens such as benzene derivatives" 

Sentence has been modified accordingly.

  1. Line 103 change "the distinguishable from others as "to "them unique as"

Changed.

  1. Line 112 change to “putting biodiversity at risk [26].

Changed.

  1. Table 1 under Lamiaceae remove or clarify which other herbs “including other herbs"

Removed.

  1. Line 133 remove "constantly"

The word constantly has been removed.

  1. Line136 remove "the health industry in areas of "

Has been removed.

  1. Line 189 separate odor and

Words have been separated.

  1. Line 204 change "in" to "of"

Has been changed.

  1. Line 208 - give name of limbo plant as it is not common

Name of plant given and changes made to sentences.

  1. Line 215 remove "In the terpenes"

Has been removed.

  1. Line 250 change "in the research in" to " on research of"

Changed as appropriate.

  1. Line 254 change "extend" to "extended"

Done.

  1. Line 259 change to "first while providing"

Done.

  1. Line 261 change "remains" to "remain"

Done.

  1. Line 264 change "also less" to "brief"

Done.

  1. Line 266 change to "In contrast, the base notes gives a perfume its"

Done.

  1. Line 267 change to “remain longer up to"

Done.

  1. Line 268 remove " the"

Done.

  1. Line 281 change to "benefits, including their pleasant aromas."

Done.

  1. Line 292 change "among" to "of"

More description of table content has been provided instead.

  1. In table 2 number 29 Juniper.... change to "pale yellow"

Changed.

  1. Table 3 linaool AND linalyl acetate change to "citrus"

Changed.

  1. Line 297 should be number 6 (not 7)

Has been renumbered correctly.

  1. Line 299 remove "progressively"

Removed.

  1. Line 301 remove "to"

Checked and found correct.

  1. Line 305 change "commonest" to "most common"

Changed.

  1. Line 337 make a table listing the 26 possible allergens

A table has been added.

  1. Line 348 change "does" to "do"

Changed.

  1. Line 352 should be "manufacturers have" and change "make sure of the" to "have"

Changed.

  1. Line 353 add "ensure product quality"

Added.

  1. Line 354 change "pass" to "divulge"

Done.

  1. Line 355 add "be advantageous"

Added.

  1. Line 362 remove "methods and"

Removed.

  1. Line 363-366 is it necessary to define what the ISO does? needs to be rewritten (and shortened) if so.

Definition has been shortened.

  1. Line 368 remove "of them" change "the maximum concentration of them" to " their maximum concentration"

Changes have been made.

  1. Line 372 change "has been “to "is"

Changed.

  1. Line 373 remove continuous

Removed.

  1. Line 375 add "dermatologic sensitivity"

Added.

  1. Line 377 change "With" to "Because of” and change to” to test specific"

Changes were made.

  1. Line 382 change "against" to "with"

Changed.

  1. Line 388 change " is diluted to double the concentration to be used regularly" to "is diluted to double the concentration," 

Changed.

  1. Line 391 should be 7.

Correction made.

  1. Line 392-428 this section needs to be rewritten - especially the grammar and spelling needs to be corrected

This section has been rechecked.

  1. Line 424 a description of NMR is not needed for the audience that reads the journal Molecules

Description was used for better comparison.

  1. Line 429-430 - need to rewrite as it is not clear

Has been properly checked and corrected.

  1. Line 458-461 needs to be rewritten for English grammar.

Sentence has been checked and rewritten.

  1. Line 458 remove "over the time"

Removed.

  1. Line 462 change "natural grade" to ""natural" labels"

Changed.

  1. Line 467 number is incorrect for the section

Corrected accordingly.

  1. Line 468-469 remove "Indeed" and remove "and do not appear to be passing so soon"

Removed.

  1. Line 475 change "apprehended as being" to "thought to be"

Changed.

  1. Line 476 change "even though are able to" to "can"

Changed.

  1. Line 484 change "watched out" to "indicated" and change "by" to "for"

Changed.

Reviewer 2 Report

Dear Authors

Essential oils are vital assets in the cosmetic industry, as, in addition to releasing pleasant aromas in different products, they are able to act as preservatives and active agents and, at the same time, offer various benefits to the skin. Furthermore, the stimulating demand for natural ingredients has contributed massively to a renewed interest of the cosmetic and wellness industries in plant derivatives, especially essential oils. This has led famous cosmetic companies to approve natural fragrances and opt for minimally processed natural ingredients, given the potential health risks associated with artificial fragrance chemicals, which are the main elements of the metic. Among the high-value essential oils used as fragrances are citrus, lavender, eucalyptus, tea tree and other floral oils, among others, while linalool, geraniol, limonene, citronellol and citral are highly valued fragrance components used in various cosmetics. Therefore, this review aimed to highlight the enormous versatility of essential oils as significant sources of natural fragrances in cosmetics and cosmeceuticals.
In recent years, given the increasing problem of antibiotic resistance, new answers to them are being sought with new therapeutic means both at systemic and topical level.
One of the main pathologies of dermatological bacterial etiology is represented by acne, and which is creating various problems with commonly used therapies that are not able to solve.
Therefore it is required to deepen this aspect as essential oils will be the new frontier in the therapy of infectious diseases. use these papers and quote them in the bibliography to carry out for this in-depth study:PMID: 32210603 ; PMID: 29908571 ; PMID: 25597924 
review English grammar and syntax necessarily.

Author Response

Reviewer #2: Revisions have been highlighted in turquoise

Essential oils are vital assets in the cosmetic industry, as, in addition to releasing pleasant aromas in different products, they are able to act as preservatives and active agents and, at the same time, offer various benefits to the skin. Furthermore, the stimulating demand for natural ingredients has contributed massively to a renewed interest of the cosmetic and wellness industries in plant derivatives, especially essential oils. This has led famous cosmetic companies to approve natural fragrances and opt for minimally processed natural ingredients, given the potential health risks associated with artificial fragrance chemicals, which are the main elements of the metic. Among the high-value essential oils used as fragrances are citrus, lavender, eucalyptus, tea tree and other floral oils, among others, while linalool, geraniol, limonene, citronellol and citral are highly valued fragrance components used in various cosmetics. Therefore, this review aimed to highlight the enormous versatility of essential oils as significant sources of natural fragrances in cosmetics and cosmeceuticals.

  1. In recent years, given the increasing problem of antibiotic resistance, new answers to them are being sought with new therapeutic means both at systemic and topical level. One of the main pathologies of dermatological bacterial etiology is represented by acne, and which is creating various problems with commonly used therapies that are not able to solve. Therefore it is required to deepen this aspect as essential oils will be the new frontier in the therapy of infectious diseases. use these papers and quote them in the bibliography to carry out for this in-depth study: PMID: 32210603 ; PMID: 29908571; PMID: 25597924.

These above mentioned references have been included in the revised paper.

  1. Review English grammar and syntax necessarily.

English grammar has been checked. Syntax and typos have been removed.

Round 2

Reviewer 1 Report

I can see the authors have improved the manuscript - there are just a few minor changes before it can be published.

line 89-90 remove "and toluene among others"

line 207 not clear what limbo is? please clarify.

the figures and tables need to be placed in a better format for the final manuscript.

Author Response

Comments from Reviewer# 2 (second round): All changes have been highlighted in green

I can see the authors have improved the manuscript - there are just a few minor changes before it can be published.

  1. Line 89-90 remove "and toluene among others"

"and toluene among others" has been removed as requested.

  1. Line 207 not clear what limbo is? please clarify.

Limbo is a plant organ. It is a gamopetalous corolla, the free portion of the corolla that forms a border at the end of the tube (Contributions from the United States National Herbarium, Volumes 50-51).

Other authors have also studied the volatiles from plants organs such as limbo, foliar sheath and rhizomes for comparing their chemical profile and biological activities. (Reference: Pharmaceutical Sciences GC/MS and GC/FID analysis and evaluation of antimicrobial performance of Aframomum sceptrum essential oils of Benin).

The word ‘’organ’’ has been added in line 207 to make it clear to readers.

  1. The figures and tables need to be placed in a better format for the final manuscript.

Figures 1 and 2 have been reformatted. In addition, during the publication process the publisher will check the quality and format tables as per their guidelines. Again we wish to thank you for sparing time to review our manuscript and granting us the chance to share it with the scientific community.